

# Marine forearc structure of eastern Java and its role in the 1994 Java tsunami earthquake

Yueyang Xia[1], Jacob Geersen[2], Dirk Klaeschen[1], Bo Ma[1], Dietrich Lange[1], Michael Riedel[1], Michael

Schnabel[3], Heidrun Kopp[1,2]

[1]GEOMAR Helmholtz Centre for Ocean Research Kiel, Wischhofstr. 1-3, 24148 Kiel, Germany

[2]Christian-Albrechts-Universität zu Kiel, Christian-Albrechts-Platz 4, 24118 Kiel, Germany

[3]Bundesanstalt für Geowissenschaften und Rohstoffe (BGR), Stilleweg 2, 30655 Hannover, Germany

*Correspondence to*: Yueyang Xia (yxia@geomar.de)

**Abstract.** We resolve a previously unrecognized shallow subducting seamount from a re-processed multichannel seismic depth

image crossing the 1994 M7.8 Java tsunami earthquake slip area. Seamount subduction is related to the uplift of the overriding

plate by lateral shortening and vertical thickening, causing pronounced back-thrusting at the landward slope of the forearc high

and the formation of splay faults branching off the landward flank of the subducting seamount. The location of the seamount

in relation to the 1994 earthquake hypocentre and its co-seismic slip model suggests that the seamount acted as a seismic

barrier to the up-dip co-seismic rupture propagation of this moderate size earthquake. The wrapping of the co-seismic slip

contours around the seamount indicates that it diverted rupture propagation, documenting the control of forearc structures on

seismic rupture.





## 1 Introduction

Tsunami earthquakes represent a special class of seismic events that rupture the very shallow portion of a subduction plate boundary (Kanamori, 1972; Satake & Tanioka, 1999). They are characterized by a longer source duration compared to
conventional earthquakes with a similar magnitude that nucleate at greater depth (e.g. Bilek & Lay, 2002). Despite being of only moderate surface wave magnitude, tsunami earthquakes commonly trigger an anomalously large tsunami. Due to the lack of severe ground shaking, coastal communities are often caught by surprise by the associated tsunami, resulting in potentially high numbers of fatalities (Satake et al., 2013). In spite of their often severe consequences, our current knowledge on tsunami earthquakes is insufficient to comprehensively understand their seismo-tectonic genesis and to identify regions that are
particularly endangered.

The reduced rupture speed, large shallow slip, and moderate shaking of earthquakes that break the shallow plate boundary might be preconditioned by a low rigidity in the outermost forearc (Bilek & Lay, 2002; Sallarès & Ranero, 2019; Şen et al., 2015). Structural features invoked to explain the unusual slow rupture of tsunami earthquakes include the presence of excess topography on the subducting plate, which may act as a localized asperity (Abercrombie et al., 2001; Tanioka et al., 1997).
Further explanations include rupture within unconsolidated subducted sediments (Kanamori, 1972; Satake & Tanioka, 1999), re-activated splay-faulting in the upper plate (Fan et al., 2017; von Huene et al., 2016; Wendt et al., 2009), vertical pop-up expulsion (Hananto et al., 2020), or inelastic shoving of unconsolidated sediments under the action of shallow slip (Seno, 2002; Tanioka & Seno, 2001).

With only 13 known events since 1896, tsunami earthquakes occur sporadic but are observed globally (Geersen, 2019). The
Java margin, which constitutes the eastern portion of the Sunda Arc (Kopp et al., 2006) was, however, affected twice by tsunami earthquakes in recent times (1994 and 2006). The 1994 Mw 7.8 earthquake (2 June, 1994 18:17:34 UTC) ruptured the shallow part of the plate boundary offshore easternmost Java (Figure 1; Abercrombie et al., 2001). Its co-seismic slip model is characterized by a non-uniform pattern, with the maximum slip under the forearc high (Abercrombie et al., 2001; Bilek & Engdahl, 2007). The induced ground motion and seafloor perturbation resulted in a severe tsunami with run-up heights of up
to ~14 meters (Tsuji et al., 1995), causing great damage to the local coastal area and approximately 250 casualties (Polet & Kanamori, 2000). The tsunami modelling for the 1994 Java earthquake reveals that the source of the larger-than-expected tsunami run-up could be linked to the horizontal displacement of the steep seafloor slope on the overriding plate (Tanioka & Satake, 1996).

The 1994 Java tsunami earthquake has been interpreted as having ruptured over a subducting seamount that induces a
localized asperity within an overall low-coupled shallow plate boundary environment (Abercrombie et al., 2001; Bilek & Engdahl, 2007). This interpretation is based on the presence of multiple seamounts within the Java trench as recognized in early side-scan data (Masson et al., 1990), the presence of a well-developed shallow forearc-high (Fig. 1b), a positive gravity anomaly under the forearc high (Fig. 1c) and the dominance of normal faulting aftershocks in the outer rise (Abercrombie et al., 2001). To date, the presence of the seamount in the peak slip region of the 1994 earthquake has not been confirmed by
marine seismic data (Lüschen et al., 2011; Shulgin et al., 2011). The previous interpretation (Abercrombie et al., 2001; Bilek



& Engdahl, 2007) is in contrast to the notion that subducting seamounts affect the plate interface as a geometrical irregularity, induce permanent brittle deformation of the overriding plate and develop a heterogeneous stress field which does not support the generation of large earthquakes (M > 8) but rather favours moderate and small events (4 < M < 8) or aseismic creep (Kodaira et al., 2000; Ruh et al., 2016; Wang & Bilek, 2011).

In this study, we image the structure of the Java margin using multichannel reflection seismic data (MCS) in the region of the 1994 tsunami earthquake in order to resolve the influence of subducting lower plate topography and upper plate structure and deformation on the seismo-tectonic genesis of the earthquake. Our study is based on enhanced processing of a multichannel seismic reflection line crossing the epicentral area. Re-processing of the profile aimed to improve the subsurface velocity model and to enhance the multiple suppression in order to augment the imaging quality. Pre-stack depth migration refines a

combined P-wave velocity model from a MCS reflection tomography (Xia et al., 2021) and ocean bottom seismometer (OBS) refraction tomography (Shulgin et al., 2011).



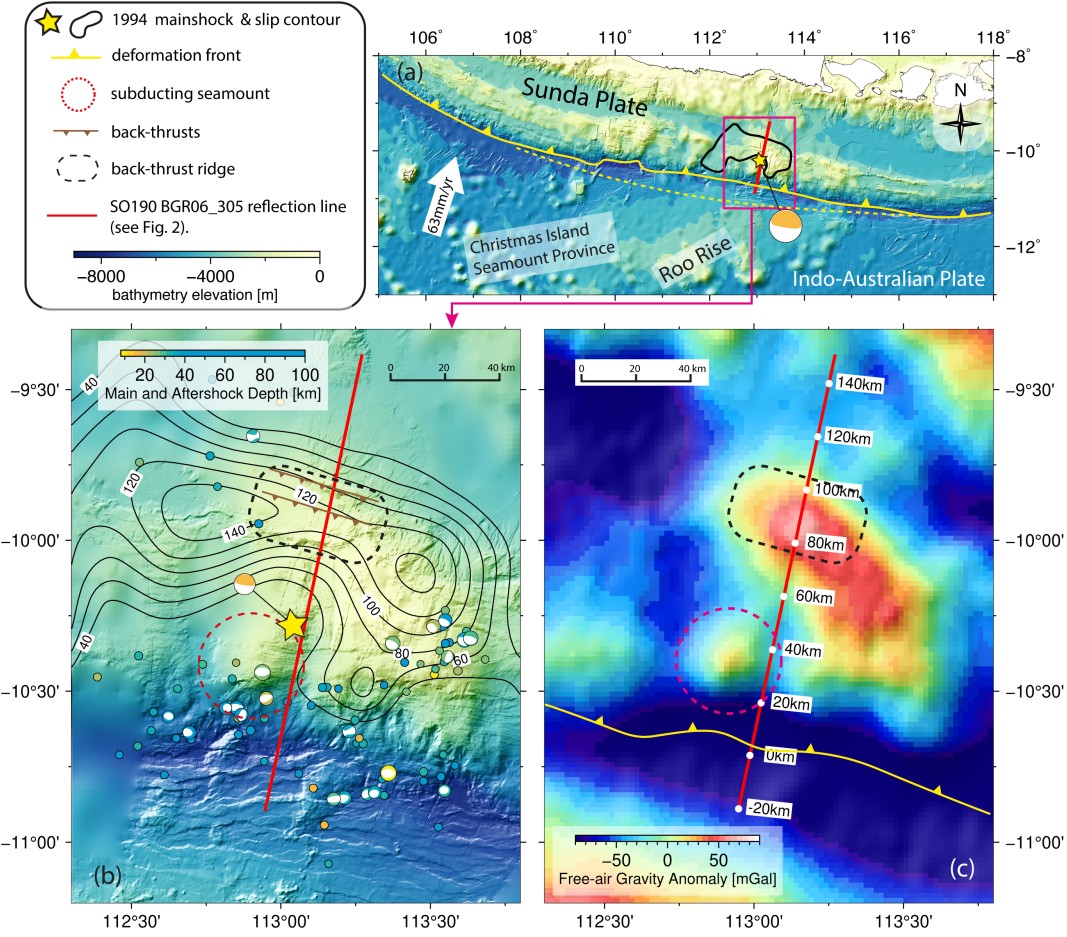


**Figure 1.** (a) Bathymetric overview, from the General Bathymetric Chart of the Oceans (GEBCO, 2020), of the Eastern Java Margin. Yellow line: deformation front. Dashed yellow line: assumed trend of the deformation front prior to frontal erosion related to the Roo Rise (Kopp et al., 2006). Yellow star: 1994 Java Tsunami earthquake epicentre. The moment tensor of the 1994 mainshock (gCMT, Dziewonski et al., 1981; Ekström et al., 2012) is plotted at the position of the 1994 epicentre from

the ISC-EHB Bulletin catalogue (Engdahl et al., 2020). Black line: approximate rupture area of the 1994 Java Tsunami earthquake (Bilek & Engdahl 2007). Red line: Seismic line SO190 BGR06_305 shown in figure 2. (b) Local bathymetry acquired during SO190 cruise overlain on the GEBCO_2020 grid. Black lines are slip contours (in cm) of the 1994 Java tsunami earthquake from Bilek & Engdahl 2007. The locations of the mainshock and largest aftershocks (timespan: 03 June - 14 October 1994) are plotted from the ISC-EHB Bulletin catalogue (Engdahl et al., 2020). Focal mechanisms are from the

gCMT catalogue (Dziewonski et al., 1981; Ekström et al., 2012). The black dashed rectangle indicates the back-thrust ridge. The red dashed circle marks the bathymetric elevation associated with the shallow subducting seamount. Note the decrease of co-seismic slip and bending of contour lines around the subducting seamount. Brown lines: back-thrust faults imaged in the seismic profile. (c) Free-air gravity anomaly (Sandwell et al., 2014).





## 2 Data and Methods

The multichannel seismic reflection profile SO190 BGR06_305 was acquired in 2006 under the scope of the Sindbad
Project during RV SONNE cruise SO190 conducted by the Federal Institute for Geosciences and Natural Resources (BGR)
(Müller & Neben, 2006). The profile is part of a 2D survey covering the marine forearc offshore eastern Java and the Lesser
Sunda Islands (Lüschen et al., 2011; Planert et al., 2010; Shulgin et al., 2011). BGR's G-Gun airgun array was used as a
seismic source with a maximum total volume of 3,100 in³ (50.8 l) and a towing depth of 6 m. Seismic signals were recorded

by the 3,000 m long digital cable of BGR's SEAL System, which consists of 20 seismic sections with 240 channels in total.

Seismic pre-processing is summarized in Table S1 and is based on a routine that includes geometry set-up, common midpoint
(CMP) binning, zero offset traces padding, bandpass filtering, shot interpolation, and random noise attenuation. We employed
a free surface-related multiple prediction method to predict the multiple waves from the primary events based on the Kirchhoff
integral (Verschuur et al., 1992) (Fig. 1). An adaptive subtraction was used to eliminate the multiple (Guitton & Verschuur,

2004) and was applied using cascaded frequency bands (Table S1, Figs. S1 and S2). Three bands of frequency (0-12 Hz, 12-
50 Hz, and 50-90 Hz) are defined in the adaptive subtraction to fit the spectrum discrepancies of the two inputs of the original
data and modelled multiple (Fig. S2). This novel multiple suppression strategy greatly improved the resolution at depth by
unveiling and preserving the deeper reflections that were previously blurred and covered by the seafloor multiple (Figs S3 and
S4). Figure S4 illustrates the efficiency of this application. The multiple overprinted on the primaries in Figure S4 (a) are step-

by-step eliminated by the adaptive subtraction, Radon dip filter, and the amplitude clipping. Remarkably, the adaptive
subtraction of modelled multiple (Fig. S4, panel c) removed most of the multiple with similar dipping angle as the primaries,
which are difficult to discriminate using a conventional dip filter (e.g., Radon filter in Fig. S4, panel d) at the near-offset. The
initial velocity analysis is performed in the time domain with a CMP increment of 250 m and converted to the depth domain.
This MCS $v_p$ model is subsequently merged with the OBS refraction model of Shulgin et al. (2011) to correct the $v_p$ field at

greater depth (2 – 4 km below seafloor), from where little effective MCS reflection signal and moveout sensitivity is recorded
(marked as the white band in Fig. S5). The merging of the velocity models is conducted with a smooth taper zone with a width
of ~ 2 km to eliminate any abrupt $v_p$ changes. We used the final merged $v_p$ model as the initial model for the pre-stack depth
migration. Subsequently, we conducted an iterative ray-based reflection depth tomography with a warping method to minimize
the residual depth error to retrieve an optimized $v_p$ model (Xia et al., 2021). Most significant are the image improvements

compared to Lüschen et al. (2011, Fig. 3) in the shallow subsurface structure of splay fault-a, b, and c (Fig. 2) and in the deeper
parts where the seafloor multiple overprinted the primary reflections.

Multibeam bathymetric data were collected during the SO190 cruise, using a SIMRAD EM120 multibeam echo sounder.
The bathymetry survey was edited and merged with the GEBCO_2020 bathymetry (GEBCO, 2020) in the areas not covered
by the multibeam soundings. Gravity data in this study are from Sandwell et al. (2014) based on satellite radar measurements.




## 3 Results

The oceanic Indo-Australian Plate off Java features a large number of seamounts and oceanic plateau (e.g. the Roo Rise) that form the northern extension of the Christmas Island Seamount Province (Fig. 1a). In the region of the 1994 earthquake, oceanic basement relief breaching the sediment infill is observed in the trench and currently colliding with the marine forearc

(Fig. 1b) (Masson et al., 1990; Kopp, 2011). The oceanic plate, which locally carries up to 1000 m of sediment, is shaped by bending related normal faults. The normal faults repeatedly offset the oceanic basement and shallow trench sediments, including the seamounts, leading to prominent seafloor escarpments (Figs 1b, 2). From around kilometre 5 landward of the trench, the décollement forms a ~40 km wide bulge or topographic elevation (Fig. 2b). The dip angle of the subducting oceanic basement increases from 2.6° on the seaward side of the bulge to ~10.2° on its landward side (Fig. 2b). We interpret the bulge

as a large subducting ridge or seamount (hereafter referred to as seamount) and find no indication for the presence of multiple small seamounts along the seismic profile, as previously suggested by Lüschen et al. (2011). With a height of ~2 km (Fig. 2) and a possible width of 40 km (interpreted from the bathymetry and free-air gravity (Fig. 1b-c)), the seamount corresponds to some of the broad and wide topographic highs observed in the seafloor bathymetry that are associated with the Christmas Island Seamount Province (Fig. 1a). The seismic reflection pattern of the plate boundary differs substantially up-dip and down-

dip of the seamount (Fig. 2a). High amplitude and negative polarity patches are imaged on the seaward side of the seamount crest (Figs 2a, 3a, kilometres: 15 – 30), and associated a low vp (2500 – 3500 m/s) in the outermost forearc (Fig. 2b, depth 6 – 8 km, kilometres: 15 – 30). On the landward side, an increased vp (4000 – 5000 m/s) is inferred from the wide-angle seismic data (Shulgin et al., 2011) at the leading edge of the seamount (Fig. 2b, kilometres: 35 – 60, depth: 8 – 12 km) followed by a slight decrease farther landward (kilometres: 60 – 70, depth: 12 – 14 km).

Below the lowermost continental slope (Fig. 2b, kilometres: 0 – 12), a distinct set of landward dipping imbricate faults with high amplitudes defines the actively deforming frontal prism. The internal structure of the middle slope regime (Fig. 2, kilometres: 12 – 32) is characterized by lower amplitudes and an overall fine-scale fragmented reflection pattern (Fig. 2a-b). Comparable imbricate faults are much less distinct underneath the middle slope (Fig. 2a-b) than underneath the frontal prism. Both the frontal prism and middle slope domain host a steep seafloor with an inclination of about 8.3° (Fig. 2a-b). A distinct

change in the slope of the seafloor at kilometre 32 defines the transition from the steeply inclined middle slope to the almost flat forearc high that extends between kilometres 32-102 (Fig. 2). The transition correlates with a prominent splay fault system that connects from the landward flank of the subducting seamount to the seafloor (Figs 2b: splay fault - a, 3a). At shallow depths (<5 km), the main splay fault divides into several branches that crop out at the seafloor between kilometres 24-30 (Figs 2b, 3a). Reversed polarity reflections, relative to the seafloor are observed along the splay fault branches (Fig. 3a, insets (iii)

and (iv)). At kilometres 40 and 52 in the forearc high, splay faults are also imaged from the seafloor to a depth of 3.5 km below the seafloor (Fig. 2a-b: splay fault - b, - c). The transition from the forearc high into the forearc basin (Fig. 2, kilometres: 95 - 105) is defined by a pronounced back-thrust (Figs 2a-b, 3b). The back-thrust dips seaward and is traced to 9 – 12 km depth, where seismic resolution diminishes, but it may well connect to the plate boundary below (Figs 2b, 3b). It offsets the shallow sediments (vertical throw of 600 m) and links to a compressional ridge at the seafloor (Figures 1b, 3b).



A distinct positive gravity anomaly outlines the forearc high (dashed rectangle in Fig. 1c). This anomaly, however, does not correlate with a subducting topographic feature, as suggested in previous studies (Abercrombie et al., 2001; Bilek & Engdahl, 2007). Projected onto the seismic line (Fig. 2b), it correlates with a prominent block of high vp (6–7 km/s) in the island arc crust that has been interpreted as a forearc backstop (Shulgin et al., 2011). The above-mentioned back-thrust evolves along the edge of this high-velocity feature (Figs 2b, 3b, 4). A smaller, circular positive gravity anomaly is visible farther up-
dip close to the deformation front (red dashed circle in Fig. 1c). This anomaly correlates to the shallow subducting seamount under the middle slope identified in the seismic line.



**Figure 2.** (a) Pre-stack depth migrated section of seismic profile SO190 BGR06_305. (b) Seismic section overlain by the vp model (based on MCS reflection tomography above 3 km depth, velocities below from Shulgin et al. (2011), see Fig. S5), our

structural interpretation, and the aftershock seismicity (from catalogue ISC-EHB Bulletin; Engdahl et al., 2020) of the 1994 Java tsunami earthquake. The hypocentre is marked as a yellow star. Coloured circles and beach-balls are aftershocks (Timespan: 03 June, 1994 to 14 Oct, 1994) from the ISC-EHB Bulletin catalogue (Engdahl et al., 2020). Focal mechanisms from the gCMT catalogue (Dziewonski et al., 1981; Ekström et al., 2012). The well-developed forearc high (75 – 100 km) results from back-thrusting above the island arc crust backstop. A subducting seamount between kilometres 5 – 45 is overlain

by upper plate splay faults. (c) Co-seismic slip model of the 1994 earthquake along the profile (Bilek & Engdahl 2007). Peak slip occurred underneath the backstop (55 – 105 km) and decreased towards the subducting seamount (5 – 45 km).





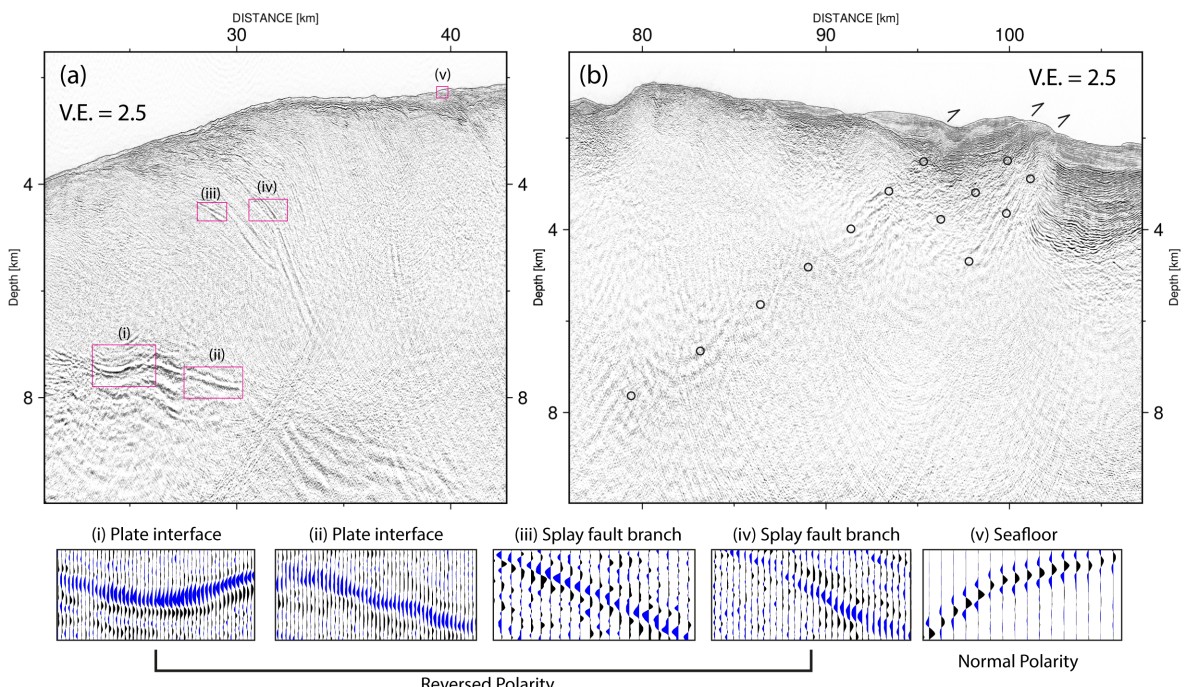

**Figure 3.** (a) Seismic section with a close-up view of the splay fault - a (compare Fig. 2b for location) branching from the landward side of the subducting seamount. Red boxes indicate close-up views shown in the lower panels. Reversed polarity reflections (relative to the seafloor) are observed at the plate interface seaward of the seamount and along the splay fault. The black and blue colours present positive and negative wavelet polarity, respectively. (b) Seismic section with a close-up view of the back-thrust (compare Fig. 2b for location). Black circles mark the back-thrust fault planes.



## 4 Discussion

The depth section of seismic line SO190 BGR06_305 (Fig. 2a), which is based on advanced seismic processing techniques, resolves the tectonic structure in the region of the 1994 Java tsunami earthquake at a level of detail that largely exceeds earlier studies (Lüschen et al., 2011; Shulgin et al., 2011). In contrast to Lüschen et al. (2011), who interpreted multiple small subducting seamounts in the shallow subduction zone, the improved imaging quality of the seismic profile reveals a single broad subducting seamount at the shallow plate boundary seaward of the forearc high. The distinct change in the dip of the

décollement from ~2.6° under the outermost forearc to >10° seaward of kilometre 40 outlines the flanks of the seamount. The seamount modulates the seafloor bathymetry, causing a small circular bathymetric elevation (red circle in Fig. 1b) and is further manifested in the circular free-air gravity anomaly close to the trench (red circle in Fig. 1c). Consistent with other well-imaged subducting seamounts (e.g. Kodaira et al., 2000; Bell et al., 2010) and results from analogue and numerical modelling (Ruh et al., 2016; Sun et al., 2020), we observe intensified compressional features at the leading edge of the seamount (e.g.,

increased vp, and enhanced splay faulting) (Fig. 2b, kilometres: 32 – 65). In contrast, gravitational relaxation (e.g., decreased vp, fine-scale fragmented internal reflection, and high plate-boundary amplitudes with reversed polarity) is observed at the trailing edge of and above the seamount (Fig. 2b, kilometres: 15 – 32). Based on the extent of the seamount (2 km high, possibly 40 km wide) and the moderate crest angle (~ 10 degrees), we speculate that the true dimension of the seamount is even larger (~ 40 – 60 km) as the seismic line might only cross the seamount's eastern flank (compare the location of the

bathymetry and gravity anomaly in Fig. 1c).

    The forearc high in the region of the 1994 Java tsunami earthquake is more evolved (shallower seafloor) compared to the adjacent regions along the margin (Fig. 1). This, in combination with the collocated gravity anomaly (dashed rectangle in Fig. 1b-c; kilometres: 75 – 105 in Fig. 2b), has fostered speculations about the presence of a subducting seamount in the peak slip region of the 1994 earthquake (e.g. Abercrombie et al., 2001). The re-processed seismic reflection image, however, suggests

that the shallow forearc high is associated with lateral shortening and vertical thickening of the upper plate ahead of a seamount currently underthrust at shallow depth. Regional uplift of the forearc slope might be enhanced by the presence of an island arc backstop (Byrne et al., 1993). The backstop underneath the forearc high is expressed as a high vp block interpreted as crystalline island arc crust due to its vp of 6 – 7 km/s (Shulgin et al., 2011). The strong lateral velocity gradient underneath the crest of the forearc high is associated with an abrupt change in material properties, manifested in back-thrusting along the well-

imaged fault plane (Fig. 3b) and the development of thrust ridges at the seafloor outcrop of the fault (Figs 1-2). Along this line of argumentation, the mature forearc high reaching shallower water depths compared to its vicinity (Figs 1-2), in the peak slip region of the 1994 Java tsunami earthquake, likely results from the combined effect of increased horizontal stress (pushing) ahead of the seamount and the presence of island arc crust serving as a rigid backstop. The resulting shortening and thickening of the upper plate are elucidated through a series of seaward vergent upper plate splay faults above the seamount and at least

two well-imaged landward vergent backthrusts along the edge of the island arc backstop.

    Abercrombie et al. (2001) and Bilek & Engdahl (2007) relocated the 1994 hypocentre and modelled the co-seismic slip. Both studies share a similar event location and a grossly similar characteristic of the co-seismic slip. The relocated hypocentre





of the 1994 earthquake and the main co-seismic slip patch are located at the leading edge of the shallow subducting seamount (Fig. 1). The co-seismic slip further seems to taper around the subducting seamount, whereas in the seamount region (red dashed circle in Figure 1b), the slip value decreases significantly (Fig. 1b, Fig. 2 b-c, kilometres: 5 – 45). These observations lead us to reconsider if the 1994 Java tsunami earthquake ruptured across a subducting seamount or if the seamount might have played a different role in the seismo-tectonic genesis of the event. From numerical models, there is evidence that subducting seamounts induce overpressures and increase shear stress at their leading edge in a region that is equivalent to their own size (Ruh et al., 2016). This would be near 1500 km$^2$ for a seamount of over 40 km in diameter as the one observed, which could be enough to generate an earthquake of M 7-8. From this, we conclude that increased shear stress in front of the subducting seamount may have preconditioned the 1994 Java tsunami earthquake.

This, however, does not solve the question of how the rupture might have evolved during the 90 seconds, which the event lasted (Abercrombie et al., 2001). The co-seismic slip models independently derived by Abercrombie et al. (2001) and by Bilek & Engdahl (2007) show similar maximum slip values around 1.5 m. This moderate co-seismic slip value makes it questionable whether the large tsunami (~14 m run-up) that widely devastated the regional shorelines (Tsuji et al., 1995) was solely generated by rupture across the shallow dipping (~8-10°) décollement. Tanioka & Satake (1996), in their approach to model the resulting tsunami through slip of a shallow dipping décollement, had to apply a mean slip value of 3.24 m. This holds true although they used a dip angle of 15° for the décollement, which significantly exceeds the 8-10° dip observed in our seismic reflection data and corresponding refraction line (Shulgin et al., 2011).

An alternative mechanism that may have contributed to the generation of the large tsunami is an activation of upper plate splay faults. In the seismic image, we observe distinct splay faults, which feature higher amplitude seismic reflections, merging at the landward side of the seamount (Figs 2b: splay fault - a, - b and - c, 3a). Generically, splay faults form when the primary fault, in this case, the plate interface, becomes critically misaligned with the principal stresses on the optimum plane (Scholz et al., 2010). Though a variety of scenarios could result in such a change of principal stress, we note that the structural modification of the plate boundary induced by the subducting seamount will cause such a misalignment of the primary stress with the basal fault and further enhances the vertical thickening and lateral shortening of the upper plate (Lallemand & Le Pichon, 1987).

Activation of the steeply dipping (e.g., maximum 40° at splay fault – a, in Fig 2) splay faults and the associated higher vertical displacement at the seafloor could potentially lead to an increase in tsunami magnitude (Wendt et al., 2009; Scholz et al., 2010). The reversed reflection polarity on the splay fault branches and shallow décollement (Fig. 3a) suggests that these splay faults are weak, likely due to high porosity and high fluid-content. The high amplitude and thus likely weak décollement (Fig. 2 a-b) on the seaward side of the subducting seamount may be stable at shallow depth (5 – 10 km) during and after the co-seismic stage. However, one may speculate if the splay faults (Fig. 2 a-b) may be activated by a moderate earthquake (Mw 6.0 – 8.0) nucleated in the conditionally stable depth range (~10 -15 km) where substantial strain can accumulate in the inter-seismic stage. While the conceptional model that the seamount may have diverted the rupture onto the well-developed upper





plate splay faults on its leading edge cannot be verified due to lacking offshore earthquake recordings, it is worth exploring because of its potential significance in magnifying tsunami amplitudes.

Wendt et al. (2009) modelled dynamic rupture scenarios with a seamount as a seismic barrier. They concluded that co-seismic rupture of steep splay faults can be introduced by a geometrical barrier like the seamount discussed here. A similar

interrelation of upper plate splay faulting and tsunami generation is discussed for the 1944 Nankai earthquake (Moore et al., 2007) and the 1946 and 1964 Alaska events, which are addressed as representing fast endmember tsunami earthquakes (Cummins & Kaneda, 2000; von Huene et al., 2016). The focal mechanism of the 1994 Java tsunami earthquake reveals slip along a shallow dipping (12°) megathrust (e.g. Abercrombie et al., 2001) and therefore does not resemble significant activation of the steeper (maximum 40°) dipping splay faults. However, one has to keep in mind that the focal mechanism only represents

the average geometry of the rupture surface. The main rupture seems to have occurred along the shallow dipping (8 – 10°) megathrust down-dip and to both sides of the seamount (compare Fig. 1). A localized activation of the splay faults above the seamount might therefore be hidden in the computed fault plane solution. The seamount, therefore, may have deflected seismic rupture onto upper plate splay faults above its leading edge, while rupture might have progressed closer to the trench to the west and east of the seamount (Fig. 4). Due to the lack of nearby seismic lines from the surrounding rupture area, the exact

structural control on the three-dimensional evolution of the rupture cannot be constrained. A similar mechanism of plate boundary rupture terminating against subducting lower plate relief is, however, discussed for the 2006 Java tsunami earthquake (Bilek & Engdahl, 2007) as well as numerous other plate boundary events (Wang & Bilek, 2011 and references therein).



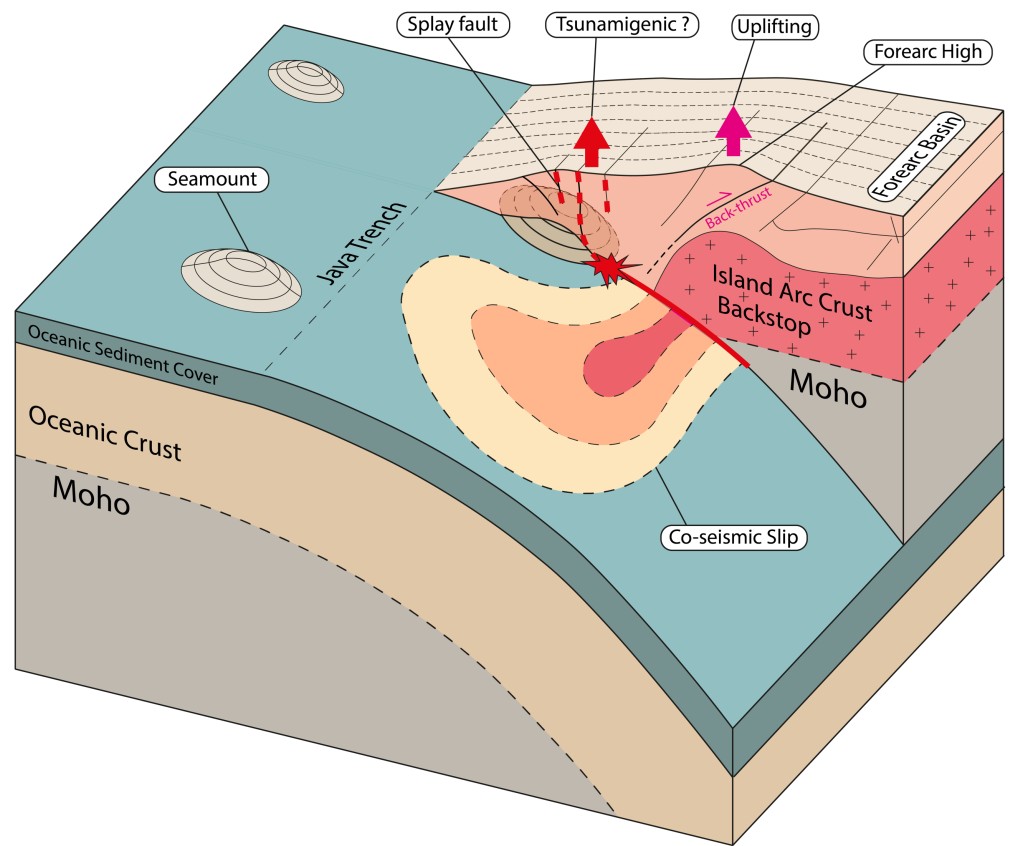


**Figure 4.** Conceptual seismo-tectonic model of the eastern Java margin in the region of the 1994 tsunami earthquake. Back-thrusting above the island arc crust backstop causes locally enhanced uplift of the forearc high. The 1994 hypocentre (red star) originated at the leading edge of a subducting seamount. The seamount stalled the co-seismic slip propagation locally along
the plate boundary.



## 5 Conclusions

A re-processed multichannel seismic reflection image with effective seafloor multiple suppression and a combined subsurface velocity from reflection and refraction tomography reveals a large subducting seamount at shallow depth (2 km –
8 km below seafloor) seaward of the rupture area of the 1994 Java tsunami earthquake. Seamount-related lateral shortening and vertical thickening of the upper plate control the uplift of the forearc high, manifested in active back-thrusting along distinct fault planes above the island arc crust backstop (Fig. 4). Furthermore, these processes favour the formation of the splay faults imaged in the seismic line. The 1994 earthquake main shock hypocentre and main co-seismic slip patch locate in front of the shallow subducting seamount. The wrapping of the co-seismic slip contours around this seamount suggests that it may
have acted as a seismic barrier during the 1994 Java tsunami earthquake (Fig. 1). These observations indicate that the seamount diverted the co-seismic rupture propagation in the up-dip direction and document the control of the shallow marine forearc structure on co-seismic rupture distribution (Fig. 4).


*Authors contributions.* YX and DK performed the computations and are responsible for the main processing. YX, JG, DK, BM, MR, MS, DL and HK helped to strengthen the overall scope and added to interpretational aspects and the discussion of the presented results. MS made the data available and was responsible for the navigation and geometry processing. YX, JG, and HK wrote the article, and all authors contributed equally to proofreading and final preparation of the manuscript.

*Data availability.* The pre-stack depth migration section of the profile BGR06_305 is available upon reasonable request. Bathymetric data from R/V SONNE cruise SO190 can be requested through the German Bundesamt für Seeschifffahrt und Hydrographie (BSH; http://www.bsh.de). Aftershock data displayed in Figures 1b and 2b are available through the ISC-EHB Bulletin catalogue (www.isc.ac.uk/isc-ehb; Engdahl et al., 2020). Focal mechanisms are available through the gCMT catalogue (https://www.globalcmt.org/; Dziewonski et al., 1981; Ekström et al., 2012). The free-air gravity data shown in Figure 1c is
available through https://topex.ucsd.edu/WWW_html/mar_grav.html (Sandwell et al., 2014)

*Competing interest.* The authors declare that they have no competing interests.

*Financial support.* R/V *SONNE* cruise SO190 and the SINDBAD project were funded by the German Federal Ministry of Education and Research (BMBF) under grants 03G0190A and 03G0190B. Y. Xia acknowledges funding from the China
Scholarship Council (grant 201506400067).





*Acknowledgements.* The seismic data were processed with Schlumberger's Omega2 seismic processing suite OMEGA and Seismic Unix - an open-source software package for seismic research and processing, Centre for Wave Phenomena, Colorado School of Mines. Bathymetry and seismic images are plotted by the Generic Mapping Tools (GMT). We thank S. Bilek, and R. Abercrombie for sharing the co-seismic slip models of the 1994 Java tsunami earthquake. We thank …. (editor and reviewers). Figure 1 was prepared with The Generic Mapping Tools (GMT).



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
