# Peer review of "Marine forearc structure of eastern Java and its role in the 1994 Java tsunami earthquake"

_Solid Earth, 2021_

## Referee Comment (RC2)

[referee-annotated manuscript omitted]

---

## Author Comment (AC1)

**Response to reviewers**

Dear Reviewers,
Dear Topical Editor Mark Allen, Executive Editor Federico Rossetti, and Editor-in-Chief CharLotte Krawczyk,

We sincerely thank both reviewers for their fair and constructive reviews, which greatly improved our manuscript. We appreciate the feedback given on the manuscript and carefully incorporated all points risen. Please find below our answers for each comment in green coloured text.

Kind regards,
Yueyang Xia on behalf of all co-authors

**Referee 1: Nathan Bangs**

1. The reprocessed seismic profile (image and velocities) is convincing in showing the subducted seamount, splay faults, and backthrust, i.e. the major structural elements involved in the processes discussed in the paper. Along with the gravity and bathymetry, the paper makes a good case for the subducted seamount. Unfortunately, the gain is low in Figure 2 and the profile is very long. Consequently, it is shrunk way down and makes it hard to see much detail. Even the enlargements in Figure 3 are very faint. It is also puzzling why magnetics data was not used to further support this interpretation (I presume it exists), but the seamount is reasonably well established.

   We appreciate the comment about the low gain of Figures 2 and 3. While the reviewer is convinced by the structural elements on which our discussion and conclusions are based, he rightly misses a better resolution of details. We feel that it is important to show the entire seismic section, but in order to highlight some of the detailed structures, we have now included an additional panel in Figure 2. Panels a and b in Figure 2 now show the main segments of the seismic profile in two parts, while the interpreted profile overlain by the velocity field is displayed in its entirety in panel c. The new graphics allow presentation of the profile over its entire length of ~170 km, while highlighting structures that are essential for the discussion and conclusions. We have also increased the gain in Figure 2. The available magnetic data does not cover the summit of the seamount. There is only one line available which was acquired at the same location as seismic line BGR06-305. In the area between the trench and 50 km landward the magnetic anomaly shows a broad plateau. Probably, the seamount is already too deep to cause an interpretable signal in the magnetic data. Details and images can be found in the cruise report of SO190 Leg 1:
   https://doi.pangaea.de/10.2312/cr_so190_1

   The cruise report of Leg 2 (refraction data) is available here:
   https://oceanrep.geomar.de/362/

2. The main concern I have with the paper is that the structures that this paper associates with the seamount and invoked to explain the changes in slip behavior are not unique to seamounts and their role here is not well tied to the seamount. Splay faults and backthrusts are very common in settings without seamounts, so the fact that they are seen on the profile is not evidence that they formed due to the seamount as is stated in the abstract (Line 15) and discussion (Line 184). With just one profile, and one that appears to be on the very far flank of the seamount (at least on the flank of the bathymetric and gravity highs), it is hard to tell how the structures and properties (Vp; Line 185) claimed to be associated with the seamount (Line 195; the profile is not well positioned to support this statement) are

associated with it, or even anomalous relative to the margin as a whole. Are these structures and properties typical along this margin in areas that are not subducting seamounts? Did they form during this seamount subduction or were they pre-existing, possibly even developed from an earlier subducted seamount?

We agree with the reviewer that backthrusts and splay faults certainly are not unique to our study area or to the Sunda/Java margin. Furthermore, the reviewer is correct in pointing out that these features did not necessarily form due to the presence of the seamount. We have revised the abstract and text accordingly to clearly distinguish between observations (backthrust, splay faults) and a causal link with the seamount:

Abstract: *Seamount subduction occurs where the overriding plate experiences uplift by lateral shortening and vertical thickening. Pronounced back-thrusting at the landward slope of the forearc high and the formation of splay faults branching off the landward flank of the subducting seamount are observed.*
Text: *Consistent with other well- imaged subducting seamounts (e.g. Kodaira et al., 2000; Bell et al., 2010) and results from analogue and numerical modelling (Ruh et al., 2016; Sun et al., 2020), we observe intensified compressional features at the leading edge of the seamount (Fig. 2b, kilometres: 32 – 65).*

We also share the reviewer's concern that the profile is not optimally positioned over the crest of the seamount, which likely is larger than seen on the seismic image (see discussion in Line 189 of the original manuscript). The entire Java margin displays highly heterogeneous features and along-strike variations are very pronounced. To this end, the reviewer in a later comment suggests to show additional data along the margin. While a number of seismic profiles exist along the margin and have been presented in previous publications by a number of groups, adding an additional 2D seismic line (or several) will not solve the problem as the profiles are spaced very far apart (mean profile distance between MCS lines is 250-300 km). Nonetheless, we include additional bathymetry data to document the along-strike variations in a regional context (see detailed explanation to comment below). The existing data along this margin (bathymetry, refraction, reflection, potential field) will not allow an analysis of the temporal evolution of structures, mainly because age data are missing, so some of the (highly relevant) questions raised by the reviewer (e.g. Did they form during this seamount subduction or were they pre-existing, possibly even developed from an earlier subducted seamount?) will remain open as they require additional dedicated data acquisition.

3. Even if these structures (splay faults and backthrusts) are a result of the seamount, are they currently active in recent earthquakes or tsunami earthquakes? Their existence does not mean they have been active recently. Do any of these thrusts offset recent slope cover strata? Are there thrust ridges on the seafloor extending along strike? Is there evidence that this transect is currently more active than regions away from the seamount? And, even if these faults have been recently active, they may not have slipped during the 1994 event as presumed here (Line 233-234). As the rupture model in Figure 1 shows, slip is downdip from the seamount and splay faults may not be involved in coseismic slip. They may slip aseismically during the interseismic period. The scenario presented here is certainly possible and intriguing, but establishing any link between the seamount, upper plate structures and slip along any specific fault in a recent earthquake is a high bar to reach and requires more data to establish very convincingly.

Nathan Bangs raises the important question whether the observed structures have been active recently. Active backthrusting of the forearc high is ongoing, as evidenced from the seafloor offsets caused by the faults (compare Figure 3 Panel b, offset of shallow sediments and seafloor by ~600 m at profile km 97, compare Line 323-235). The imaged splay faults similarly affect the most recent seafloor sediment drape and partially offset the seafloor, indicating recent activity (Figure 2, panel a, around profile km 39.5-41 and 51-55. Figure 3, panel a, around profile km 39.5 and 35-36). Further observations are included in the manuscript (Line 157: the main splay fault divides into several branches that crop out at the seafloor between kilometres 24-30 (Figs 2a, 2c, 3a). Line 215: Splay fault -b (Fig. 2a) causes a minor seafloor offset in the seismic section, while splay fault -c offsets the seafloor by ~500 m as seen both in the seismic section (Fig. 2a) and bathymetry map (Fig. 4b), indicating recent activity.).

It remains, however, of course unresolved if fault activity occurs during the coseismic phase, or possibly aseismically. As pointed out by the reviewer, both scenarios may be possible, but with the available data one may not differentiate between them. Assessing the link between the splay fault and the coseismic activity is, however, not solely limited by the lack of marine geophysical images from the margin segment, but also the lack of local off-shore geodesy observations. Similarly, it is impossible to distinguish whether this transect is more active than other regions along the margin.

Accordingly, we have adjusted some statements in the manuscript:

Line 18-20 in original manuscript: *The wrapping of the co-seismic slip contours around the seamount indicates that it diverted rupture propagation, documenting the control of forearc structures on seismic rupture.* → This sentence is deleted from the manuscript in order to avoid the link between the seamount, upper plate structures and slip along any specific fault in a recent earthquake.

Line 67: *In this study, we image the structure of the Java margin using multichannel reflection seismic data (MCS) in the region of the 1994 tsunami earthquake in order to resolve the relation of subducting lower plate topography and upper plate structure to the co-seismic slip distribution.*

4.  Finally, the discussion on splay faults, subducted seamounts and tsunami magnitudes (lines 233-242) is extremely speculative. The stresses related to the seamount, the relative strengths of the faults inferred from reflection amplitudes, shears stresses along faults, etc. are not constrained well enough to make this kind of assessment. The scenario involving splay faults that the authors describe is possible without a seamount (there is no seamount involved in the Nankai case referenced: Moore et al., 2007; Line 245). The question is whether the seamount is enhancing this process somehow, yet there is no evidence presented that it has.

This comment summarizes the main concern of the reviewer. Our intention here was to put forward a conceptual model, but as we state in the manuscript (Line 240-242 in original version) the lack of offshore earthquake recordings hinders a detailed exploration of this scenario. We thus meet the reviewer's concern and have deleted this part of the discussion and related statements in the conclusions:

Line 233-242 in original: this paragraph (except Line 235 on reversed polarity) is deleted from the manuscript in order to avoid the notion that the splay faults are generated or activated during the 1994 event.
Line 279: *Furthermore, these processes favour the formation of the splay faults imaged in the seismic line.* → this statement is deleted from the manuscript.

In addition, we have slightly modified Figure 4 and deleted the arrow indicating the tsunamigenic motion of the splay faults during the co-seismic phase.

---

## Author Comment (AC2)

**Response to reviewers**

Dear Reviewers,
Dear Topical Editor Mark Allen, Executive Editor Federico Rossetti, and Editor-in-Chief CharLotte Krawczyk,

We sincerely thank both reviewers for their fair and constructive reviews, which greatly improved our manuscript. We appreciate the feedback given on the manuscript and carefully incorporated all points risen. Please find below our answers for each comment in green coloured text.

Kind regards,
Yueyang Xia on behalf of all co-authors

**Referee 2: Sara Martínez-Loriente**

1. In my opinion, the presence of the subducting seamount, splay faults and back-thrusts is well resolved. My main concern is related to the lack of more evidence showing the lateral extension of these structures and thus confirming the link proposed by the authors between the presence of the seamount and the splay faults and back-thrusts, as well as their role during the seismic event. In addition, the authors claim to the physical properties related to the presence of this elements interpreted on the profile to justify their interpretation, but they are hard to see to me in this profile. Are these same structures seen elsewhere along the margin? Is its role in the sismogenesis process the same?

   This comment raises very similar concerns to the first reviewer. We take this remark very seriously and have adjusted the text accordingly. Please refer to Comment #2 of Reviewer 1 for details on how we adjusted the abstract and main text.

2. Finally, in my opinion the last part of the discussion section is quite speculative, where the authors propose a possible activation of the splay faults (L238-240; L247-254). I do not see any evidence (eg, in the seismic profile or in the seismic Vp structure) of the recent activity of the splay faults (deformation of the most recent sediments) as the authors state in L225.

   This concern was also voiced by Nathan Bangs (reviewer 1) – see his comments 3 and 4 and our response. We have deleted the paragraph on the possible activation scenario (Lines 233-242 in original manuscript) and have revised the text of Lines 247-254, where we now focus on the slip distribution and have deleted references and comments on a conceptual model involving the formation of splay faults due to seamount subduction or their activation during the co-seismic phase:
   *Yang et al. (2012, 2013) modelled a dynamic rupture scenario with a seamount as a seismic barrier. The seamount imaged on our seismic profile may have halted seismic rupture at its leading edge, while rupture might have progressed closer to the trench to the west and east of the seamount (Fig. 5). Due to the lack of 3D seismic coverage of the rupture area, the exact structural control on the three-dimensional evolution of the rupture cannot be constrained. A similar mechanism of plate boundary rupture terminating against subducting lower plate relief is, however, discussed for the 2006 Java tsunami earthquake (Bilek & Engdahl, 2007) as well as numerous other plate boundary events (Wang & Bilek, 2011 and references therein).*
   While we provide evidence for recent fault activity (Line 157: the main splay fault divides into several branches that crop out at the seafloor between kilometres 24-30 (Figs 2a, 2c,

3a). Line 215: Splay fault -b (Fig. 2a) causes a minor seafloor offset in the seismic section, while splay fault -c offsets the seafloor by ~500 m as seen both in the seismic section (Fig. 2a) and bathymetry map (Fig. 4b), indicating recent activity.), we agree with the reviewer that it remains unresolved if this fault activity is linked to megathrust earthquake rupture.

3. Minor comments
   Figures:
   I recommend doing some close-up of figure 2 to be able to observe some of the descriptions that are made in the text.
   We have split panel a in Figure 2 into two segments in order to show close-ups of the seismic data and the described features.
   I recommend incorporating the Vp contours to figure 2, without which it is impossible to observe some of the characteristics described by the authors.
   We follow the suggestion by the reviewer and have incorporated vp contours in Figure 2.
   I recommend making the back-thrust label more visible in figure 1b, as they are difficult to distinguish.
   We have edited Figure 1b accordingly.

   In addition to these edits, we have incorporated all annotations in the provided pdf version by reviewer 2:
   - Additional reference Martinez-Loriente et al., 2019 added
   - Back-thrusts enhanced in Figure 1
   - Interpretation of single seamount instead of multiple small ones moved from results to discussion section
   - Close-ups of seismic section added to Figure 2
   - Vp contours added to Figure 2
   - Comparison to region west of seismic line added to discussion section
   - Section on activation of splay faults deleted
   - Speculation on role of splay faults removed